# Global burden of polycystic ovary syndrome in women of reproductive age, 1990–2021: Analysis of the global burden of disease study 2021 with projections to 2050

Xinkuo Zheng[1]☯*, Meishen Liu[2], Zhaohui Bai[3], Ling Wu[1], Lili Geng[4], Yun Shen[4], Jing Na[5], Haonan Qiu[6], Yalin Xi[1]☯*

**1** Department of Pharmacy, Central Hospital of Dalian University of Technology, Dalian, Liaoning, China, **2** Department of Pharmacy, The Second Affiliated Hospital of Dalian Medical University, Dalian, Liaoning, China, **3** Department of Pharmacy Administration and Clinical Pharmacy, School of Pharmaceutical Sciences, Peking University, Beijing, China, **4** Department of Infectious Diseases, Central Hospital of Dalian University of Technology, Dalian, Liaoning, China, **5** Department of Gynecology and Obstetrics, The Second Affiliated Hospital of Dalian Medical University, Dalian, Liaoning, China, **6** Department of Colorectal Surgery, Liaoning Cancer Institute and Hospital, Shenyang, Liaoning, China

☯ These authors contributed equally to this work.
* xinkuo1994@hotmail.com (XZ); yalinxi@163.com (YX)

## Abstract

### Objective

To evaluate the global burden of polycystic ovary syndrome (PCOS) among women of reproductive age from 1990 to 2021.

### Methods

This study analyzed population-based data from the Global Burden of Disease (GBD) database on the incidence, prevalence, and disability-adjusted life years (DALYs) of PCOS in reproductive-aged women. Age-standardized rates for incidence (ASIR), prevalence (ASPR), and DALYs (ASDR) were calculated per 100,000 population, with 95% uncertainty intervals (UIs). Data from 204 countries and territories were stratified by age, location, and socio-demographic index (SDI).

### Results

Globally, the burden of PCOS among reproductive-aged women in 2021 was characterized by ASIR of 64.44 (95% UI: 39.07, 103.40) per 100,000 population, ASPR of 3364.53 (95% UI: 2395.08, 4681.81) per 100,000 population, and ASDR of 29.51 (95% UI: 13.09, 61.49) per 100,000 population. Moreover, the pace of increase in ASIR, ASPR, and ASDR accelerated during 1990–2021, with estimated annual percentage change (EAPC) of 0.65 (95% CI: 0.62, 0.69), 0.74 (95% CI: 0.70, 0.77), and 0.72 (95% CI: 0.68, 0.76), respectively. Among the five SDI regions, the middle SDI

**Data availability statement:** The data utilized in these analyses are publicly accessible through the GBD Results Tool, available at https://vizhub.healthdata.org/gbd-results/. All raw data utilized in this study are publicly accessible on GitHub at: https://github.com/Xinkuo-oh/GBD2021-Polycystic-ovary-syndrome-Datasets.git.

**Funding:** The author(s) received no specific funding for this work.

**Competing interests:** The authors have declared that no competing interests exist.

region exhibited the highest EAPCs for incidence (1.39; 95% CI: 1.34, 1.43), prevalence (1.39; 95% CI: 1.34, 1.43), and DALYs (1.73; 95% CI: 1.69, 1.78). Regionally, the High-income Asia Pacific region demonstrated the highest ASIR of 308.16 (95% UI: 485.83, 171.53) per 100,000 population. At the national level, Japan exhibited the highest ASIR of 360.92 (95% UI: 199.08, 573.59) per 100,000 population.

## Conclusion

The global burden of PCOS among women of reproductive age has shown a consistent upward trend in incidence, prevalence, and DALYs. A comprehensive understanding of the epidemiology of PCOS among women of reproductive age is essential for informing targeted prevention strategies and optimizing disease control measures.

## Introduction

PCOS is a common endocrine and metabolic disorder that predominantly affects women of reproductive age [1,2]. The global prevalence of PCOS ranges from 4% to 21%, depending on the diagnostic criteria applied [3]. The three most widely used diagnostic criteria include the NIH criteria, the Rotterdam criteria, and the Androgen Excess and PCOS Society (AE-PCOS) criteria [3]. This prevalent condition, which afflicts a significant proportion of women, represents a substantial global burden, adversely impacting physical health and quality of life at the individual level and imposing significant economic costs on healthcare systems and society [4,5]. Women of reproductive age with PCOS are at a high risk of anovulatory infertility and adverse pregnancy outcomes [6]. Additionally, PCOS exerts significant effects on multiple systems, including the cardiovascular, nervous, and endocrine-metabolic systems [7].

With rapid economic development and escalating levels of stress in daily life, risk factors associated with PCOS, such as obesity and chronic psychological stress, are increasingly affecting a significant proportion of women [8]. The rising prevalence of PCOS in many regions worldwide has led to an increasing burden on healthcare systems [9]. Therefore, a detailed assessment of the epidemiological status of PCOS is essential for comprehensive planning and optimization of healthcare resource allocation.

To conduct robust epidemiological research, it is essential to utilize comprehensive and current databases. The GBD 2021 study offers the most extensive and up-to-date estimates of PCOS burden worldwide [10]. Currently, the majority of epidemiological studies investigating the global burden of PCOS rely on data from 2019 or earlier, with many studies constrained by their focus on specific national or regional populations [5,9,11–15].

Despite the significant health impact of PCOS among women of reproductive age, there is a notable lack of comprehensive quantitative analyses evaluating its global burden over the past 31 years and projecting future trends—critical data necessary for evidence-based health policy development. Leveraging data from GBD 2021, this

study systematically investigates the burden trends of PCOS among women of reproductive age across diverse global regions. Specifically, we aim to (1) characterize the current epidemiological patterns and regional variations of PCOS, (2) quantify the impact of demographic and epidemiological determinants on PCOS burden trends over the past 31 years, and (3) develop data-driven projections of PCOS burden through 2050 using advanced modeling approaches.

## Materials and methods

### Overview and methodological details

The GBD study, widely regarded as the most comprehensive and systematic initiative in global epidemiological research, is coordinated by the Institute for Health Metrics and Evaluation (IHME) at the University of Washington. This landmark initiative provides rigorous, standardized estimates of health loss attributable to a comprehensive spectrum of diseases, injuries, and risk factors across populations and geographies [10]. The GBD framework provides a standardized platform for cross-national and regional comparative analyses of epidemiological metrics, including incidence and mortality rates, across diverse populations and geographical settings [16]. The GBD methodology quantifies disease burden through several principal metrics: incidence, prevalence, death and DALYs, where DALYs represent a composite measure integrating years of life lost (YLL) from premature mortality and years lived with disability (YLD).

In this GBD-based analysis, the burden of PCOS was quantified using three core metrics: incidence, prevalence, and DALYs. Women of reproductive age (15–49 years) were selected as the primary study population from 1990 to 2021, with participants stratified into 5-year age groups: 15–19 years, 20–24 years, 25–29 years, 30–34 years, 35–39 years, 40–44 years, and 45–49 years. Data pertaining to PCOS in this study were obtained from the Global Health Data Exchange (GHDx) and its affiliated tools. On February 8, 2025, data were retrieved from GBD 2021 for research analysis, with no personally identifiable information accessible to the researchers at any stage of the study. For further reference, the GBD Results Tool is available at https://vizhub.healthdata.org/gbd-results/. However, due to the absence of data on race, underlying diseases, and other relevant patient characteristics in the GBD database, these parameters were not included in our analysis. As the GBD database is publicly accessible, this study was exempt from formal ethical review.

### Socio-demographic index

The SDI, a composite measure developed to capture the socio-economic determinants influencing population health outcomes, is derived from the geometric mean of three normalized indices: total fertility rate among individuals under 25 years, average educational attainment in populations aged 15 years and older, and lag-distributed income per capita [17]. Countries and territories were stratified into five distinct socio-development levels according to predefined SDI thresholds: low [0–0.4658), low-middle [0.4658–0.6188), middle [0.6188–0.7120), high-middle [0.7120–0.8103), and high [0.8103–1.0000) [18]. This stratification facilitates a systematic evaluation of the association between socioeconomic development levels and population health outcomes.

### Statistical analysis

Both the age-standardized rates (ASRs) and numbers of incidence, prevalence, and DALYs associated with PCOS among women of reproductive age, along with their corresponding 95% uncertainty intervals, were estimated per 100,000 population based on data from GBD database. The APC and associated 95% CIs were calculated by fitting regression models to assess temporal trends within each segmented time period [19]. Additionally, the average annual percent change (AAPC) and corresponding 95% CIs were derived to evaluate the overall trend across the entire study period from 1990 to 2021 [19]. The EAPC and its corresponding 95% CIs were derived by fitting log-linear regression models to assess temporal trends in ASIR, ASPR, and ASDR over the period from 1990 to 2021. The BAPC model, which incorporates the assumption of smooth temporal variations in age, period, and cohort effects while assigning appropriate prior distributions to all

unknown parameters, enables robust and biologically plausible predictions [20,21]. Leveraging this approach, we applied the BAPC model to project incidence, prevalence, and DALYs for the year 2050. Fitted regression curves were employed to examine the association between disease burden metrics and the SDI. Meanwhile, to evaluate the association between the burden of PCOS and SDI levels, we conducted frontier analysis to develop an ASDR-based frontier model using SDI as the primary predictor [22].

All statistical analyses and visualizations were performed using R software (version 4.4.2) and JD_GBDR (V2.36, Jingding Medical Technology Co., Ltd). A two-sided p-value threshold of <0.05 was adopted to determine statistical significance.

## Result

### Global burden trends of PCOS among women of reproductive age

**Incidence.** Large-scale data analyses have revealed notable fluctuations in the global epidemiological trends of PCOS among women of reproductive age, with a clear upward trajectory in the incidence rate. The most pronounced age-standardized APC was observed from 2016 to 2021, amounting to 1.15% (95%CI: 1.08, 1.22) (Fig 1A). Furthermore, the peak ASIR reached 64.44 (95%UI: 39.07, 103.40) per 100,000 population in 2021 (Table 1). Between 1990 and 2021, the global incidence cases of PCOS increased from $786.95*10^3$ (95%UI: $470.50*10^3$, $1290.21*10^3$) to $1175.07*10^3$ (95%UI: $711.34*10^3$, $1887.25*10^3$), corresponding to a 47.03% rise over this 31-year period (Table 1). Similarly, the ASIR increased from 52.00 (95%UI: 31.02, 85.27) per 100,000 population in 1990 to 64.44 (95%UI: 39.07, 103.40) per 100,000 population in 2021, an overall increase of 23.92% (Table 1, Figs 2A and 2B). The EAPC was 0.65 (95% CI: 0.62, 0.69) (Table 1, Fig 2C). Notably, the incidence of PCOS in women showed an increasing trend in the age groups of 15–19, 30–34, 35–39, 40–44, and 45–49 years, with the most substantial increase observed in the 15–19 age group (27.00%). Conversely, a decreasing trend was observed in the 20–24 and 25–29 age groups, with the most significant decline occurring in the 20–24 age group (1.75%) (S1 Fig).

**Prevalence.** Aligned with the observed incidence patterns, the prevalence rate of PCOS showed a sustained increase throughout the past three decades. The most pronounced age-standardized APC was observed from 2016 to 2021, amounting to 1.15% (95%CI: 1.05, 1.24) (Fig 1B). Furthermore, the peak ASPR reached 3364.53 (95%UI: 2395.08, 4681.81) per 100,000 population in 2021 (S1 Table). Between 1990 and 2021, the global prevalence cases of PCOS increased from $34806.51*10^3$ (95%UI: $24730.06*10^3$, $48620.13*10^3$) to $65767.55*10^3$ (95%UI: $46839.86*10^3$, $91498.22*10^3$), corresponding to an 88.95% rise over this 31-year period (S1 Table). In parallel, the ASPR increased from 2628.48 (95%UI: 1870.21, 3668.82) per 100,000 population in 1990 to 3364.53 (95%UI: 2395.08, 4681.81) per 100,000 population in 2021, an overall increase of 28.00% (S1 Table, S2A and S2B Fig). The EAPC was 0.74 (95% CI: 0.70, 0.77) (S2C Fig). Of particular note, the prevalence increased in all age segments, with the largest increase in women aged 20–24 (32.82%) and the smallest increase in women aged 45–49 (22.97%) (S3 Fig).

**DALYs.** Consistent with the observed trends in both incidence and prevalence, PCOS-related DALYs exhibited a continuous increase throughout the three-decade observation period. The most pronounced age-standardized APC was observed from 2015 to 2021, amounting to 1.13% (95%CI: 1.04, 1.21) (Fig 1C). Furthermore, the peak ASDR reached 29.51 (95%UI: 13.09, 61.49) per 100,000 population in 2021 (S2 Table). Between 1990 and 2021, the global DALYs of PCOS increased from $307.94*10^3$ (95%UI: $136.41*10^3$, $644.50*10^3$) to $576.05*10^3$ (95%UI: $255.58*10^3$, $1200.18*10^3$), corresponding to an 87.07% rise over this 31-year period. Correspondingly, the ASDR increased from 23.16 (95%UI: 10.27, 48.43) per 100,000 population in 1990 to 29.51 (95%UI: 13.09, 61.49) per 100,000 population in 2021, an overall increase of 27.42% (S2 Table, S4A and S4B Fig). The EAPC was 0.72 (95% CI: 0.68, 0.76) (S4C Fig). Remarkably, the DALYs increased in all age segments, with the largest increase in women aged 20–24 (32.18%) and the smallest increase in women aged 45–49 (22.97%) (S5 Fig).

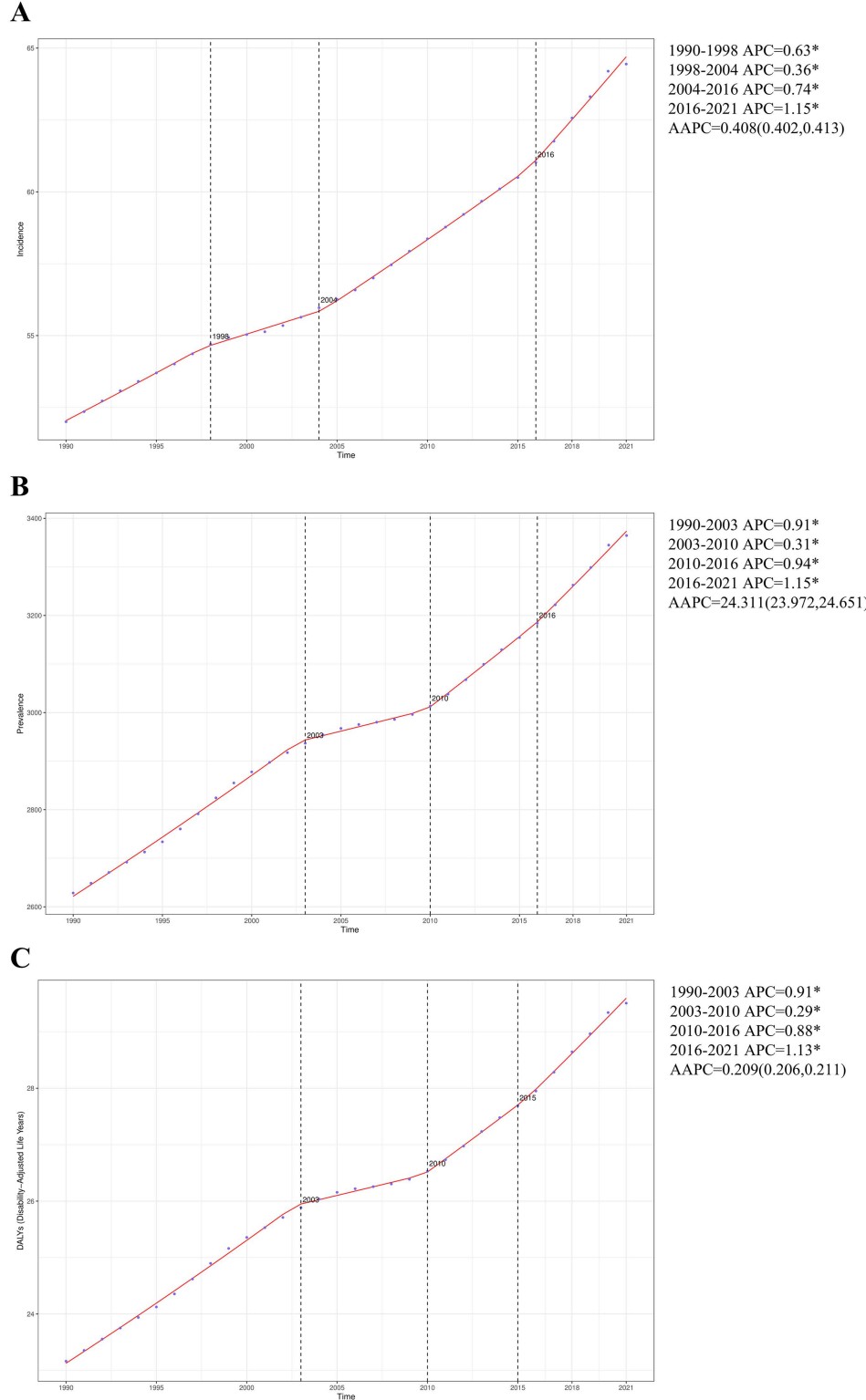

**Fig 1. APC and trends in the incidence, prevalence, and DALYs among global reproductive-aged women from 1990 to 2021.** (A) ASIR. (B) ASPR. (C) ASDR.

**Table 1. Incidence of PCOS among women of reproductive age between 1990 and 2021 at the global and regional level.**

| Location | 1990 | | 2021 | | 1990–2021 |
|---|---|---|---|---|---|
| | All-age cases n*10³ (95% UI) | ASIR per 100,000 population n (95% UI) | All-age cases n*10³ (95% UI) | ASIR per 100,000 population n (95% UI) | EAPC n (95% CI) |
| Global | 786.95 (470.50, 1290.21) | 52.00 (31.02, 85.27) | 1175.07 (711.34, 1887.25) | 64.44 (39.07, 103.40) | 0.65 (0.62, 0.69) |
| SDI | | | | | |
| Low | 34.47 (20.06, 58.40) | 24.23 (13.93, 41.20) | 110.65 (65.26, 184.41) | 31.35 (18.29, 52.50) | 0.92 (0.89, 0.94) |
| Low-middle | 110.61 (66.08, 183.25) | 32.66 (19.35, 54.27) | 250.78 (149.41, 413.99) | 46.65 (27.76, 76.97) | 1.28 (1.23, 1.32) |
| Middle | 254.93 (155.24, 412.63) | 47.11 (28.56, 76.29) | 378.02 (231.67, 605.40) | 71.14 (43.76, 113.70) | 1.39 (1.34, 1.43) |
| High-middle | 134.12 (82.01, 218.12) | 47.21 (28.89, 76.62) | 144.65 (88.41, 234.83) | 68.27 (42.10, 110.41) | 1.37 (1.30, 1.45) |
| High | 252.32 (146.41, 422.96) | 131.79 (76.61, 220.63) | 290.18 (173.36, 458.83) | 165.70 (99.24, 261.46) | 0.21 (−0.05, 0.46) |
| Regions | | | | | |
| East Asia | 139.85 (81.16, 238.97) | 37.11 (21.51, 63.15) | 135.52 (79.63, 225.68) | 60.34 (35.86, 100.15) | 1.59 (1.44, 1.73) |
| Southeast Asia | 100.07 (58.06, 165.41) | 68.73 (39.76, 113.73) | 185.10 (111.57, 301.46) | 111.59 (67.37, 181.50) | 1.89 (1.79, 1.99) |
| Oceania | 0.96 (0.59, 1.55) | 49.52 (30.11, 80.00) | 2.42 (1.51, 3.94) | 65.58 (40.69, 106.64) | 0.75 (0.61, 0.88) |
| Central Asia | 2.84 (1.62, 4.85) | 14.85 (8.45, 25.44) | 3.99 (2.27, 6.78) | 19.11 (11.02, 32.34) | 0.88 (0.82, 0.94) |
| Central Europe | 2.25 (1.28, 3.98) | 7.81 (4.43, 13.80) | 1.60 (0.92, 2.73) | 8.76 (5.19, 14.72) | 0.19 (0.07, 0.32) |
| Eastern Europe | 5.78 (3.09, 9.88) | 11.63 (6.29, 19.93) | 5.00 (2.68, 8.57) | 14.20 (7.83, 24.39) | 0.75 (0.72, 0.78) |
| High-income Asia Pacific | 109.06 (58.56, 182.37) | 247.59 (132.96, 414.08) | 75.03 (41.68, 118.51) | 308.16 (171.53, 485.83) | 0.66 (0.59, 0.74) |
| Australasia | 9.33 (5.58, 15.31) | 188.67 (112.83, 309.39) | 11.59 (6.79, 19.26) | 219.55 (128.92, 364.51) | 0.36 (0.29, 0.42) |
| Western Europe | 89.46 (59.78, 140.46) | 111.43 (74.68, 174.48) | 84.44 (55.44, 133.02) | 122.55 (80.73, 192.67) | 0.21 (0.19, 0.24) |
| Southern Latin America | 7.24 (4.18, 12.05) | 54.72 (31.52, 91.18) | 12.56 (7.30, 20.81) | 84.61 (49.37, 139.92) | 1.42 (1.22, 1.63) |
| High-income North America | 73.19 (42.24, 127.43) | 124.46 (72.03, 216.34) | 131.67 (77.04, 207.54) | 188.01 (110.12, 296.07) | −0.24 (−0.89, 0.42) |
| Caribbean | 4.34 (2.70, 7.10) | 39.77 (24.62, 65.09) | 5.37 (3.33, 8.71) | 48.07 (29.89, 77.78) | 0.68 (0.63, 0.72) |
| Andean Latin America | 9.22 (5.89, 15.48) | 75.60 (48.07, 127.04) | 15.44 (10.14, 24.69) | 95.75 (62.97, 152.74) | 0.74 (0.64, 0.83) |
| Central Latin America | 40.49 (27.38, 61.13) | 74.26 (49.99, 112.47) | 51.39 (35.03, 77.12) | 79.38 (54.19, 118.96) | −0.12 (−0.27, 0.02) |
| Tropical Latin America | 10.70 (6.47, 18.52) | 23.34 (14.00, 40.42) | 12.23 (7.28, 20.72) | 24.04 (14.51, 40.50) | −0.33 (−0.54, −0.12) |
| North Africa and Middle East | 58.63 (35.87, 96.21) | 56.62 (34.41, 93.23) | 108.45 (67.46, 177.46) | 70.52 (43.90, 115.37) | 0.83 (0.76, 0.90) |
| South Asia | 85.57 (51.22, 141.65) | 28.23 (16.75, 46.82) | 220.59 (130.24, 364.11) | 43.35 (25.60, 71.42) | 1.64 (1.53, 1.75) |

*(Continued)*

**Table 1.** (Continued)

| Location | 1990 | | 2021 | | 1990–2021 |
|---|---|---|---|---|---|
| | All-age cases n*10³ (95% UI) | ASIR per 100,000 population n (95% UI) | All-age cases n*10³ (95% UI) | ASIR per 100,000 population n (95% UI) | EAPC n (95% CI) |
| Central Sub-Saharan Africa | 3.39 (1.92, 5.83) | 21.12 (11.80, 36.55) | 12.22 (7.17, 20.62) | 28.80 (16.68, 48.90) | 0.96 (0.84, 1.08) |
| Eastern Sub-Saharan Africa | 14.33 (8.23, 24.47) | 24.80 (14.05, 42.50) | 41.69 (24.20, 70.45) | 29.75 (17.07, 50.50) | 0.64 (0.61, 0.67) |
| Southern Sub-Saharan Africa | 6.29 (3.69, 10.58) | 36.93 (21.50, 62.33) | 9.50 (5.56, 15.95) | 44.21 (25.89, 74.23) | 0.56 (0.49, 0.63) |
| Western Sub-Saharan Africa | 13.95 (8.06, 23.77) | 24.25 (13.83, 41.48) | 49.28 (28.62, 83.31) | 31.28 (17.97, 53.14) | 0.64 (0.51, 0.78) |

Abbreviations: ASIR, age standardized incidence rate; EAPC, estimated annual percentage change; SDI, socio-demographic index; UI, uncertainty interval; CI, confidence interval.

### Regional burden trends by SDI of PCOS among women of reproductive age

From 1990 to 2021, an overall upward trend in ASIR, ASPR, and ASDR was observed across all SDI regions, including low, low-middle, middle, high-middle, and high SDI categories (Table 1, Fig 3, S1 and S2 Table, S6 and S7 Fig). The most substantial increases were found in the number of incidence cases (250.78*103; 95% UI: 149.41*103, 413.99*103), prevalence cases (23243.05*103; 95% UI: 16338.14*103, 32445.09*103), and DALYs (203.16*103; 95% UI: 88.68*103, 427.66*103). Moreover, the incidence-associated EAPC (1.39; 95%CI: 1.34, 1.43), prevalence-associated EAPC (1.39; 95%CI: 1.34, 1.43), and DALYs-associated EAPC (1.73; 95%CI: 1.69, 1.78) of PCOS among women of reproductive age were highest in the middle SDI region (Table 1, S1 and S2 Table). The EAPC initially increases and then decreases with the rise in ASRs (ASIR, ASPR and ASDR), and the EAPC exhibits a similar trend in relation to SDI in 2021(S8 Fig). Significantly, the ASIR in high-income Asia Pacific was higher than expected, whereas Central Europe and Eastern Europe exhibited lower ASIR levels than anticipated (Fig 4). The aforementioned phenomena can also be identified in the SDI-related trends of ASPR and ASDR (S9 and S10 Fig). Besides, significant variations are observed across medium SDI regions, with some regions consistently remaining below expected levels from 1990 to 2021, while others exceed expected levels during the same period (Fig 4, S9 and S10 Fig).

### National burden trends of PCOS among women of reproductive age

**Incidence.** In 2021, India reported the highest absolute number of PCOS cases globally, with an estimated 180.50*10³ cases (95% UI: 107.45*10³, 296.81*10³). Conversely, Japan demonstrated the highest ASIR at 360.92 (95% UI: 199.08, 573.59) per 100,000 population. Longitudinal analysis from 1990 to 2021 revealed that the Republic of Maldives experienced the most substantial increase in ASIR, with an EAPC of 2.86 (95% CI: 2.61, 3.11). In contrast, the Republic of Poland showed the most significant decrease in ASIR, with an EAPC of −0.91 (95% CI: −1.16, −0.66). Globally, the ASIR of PCOS among women of reproductive age was estimated at 64.44 (95% UI: 39.07, 103.40) per 100,000 population in 2021. This global ASIR exceeded that of 109 out of 204 countries, while remaining lower than the incidence rates observed in 95 countries (S3 Table).

**Prevalence.** In 2021, People's Republic of China reported the highest absolute number of PCOS cases globally, with an estimated 9481.52*10³ cases (95% UI: 6591.67*10³, 13488.89*10³). Conversely, Republic of Italy demonstrated the highest ASIR at 3364.53 (95%UI: 2395.08, 4681.81) per 100,000 population. Longitudinal analysis from 1990 to 2021 revealed that the Republic of Maldives experienced the most substantial increase in ASPR, with an EAPC of 3.39 (95%

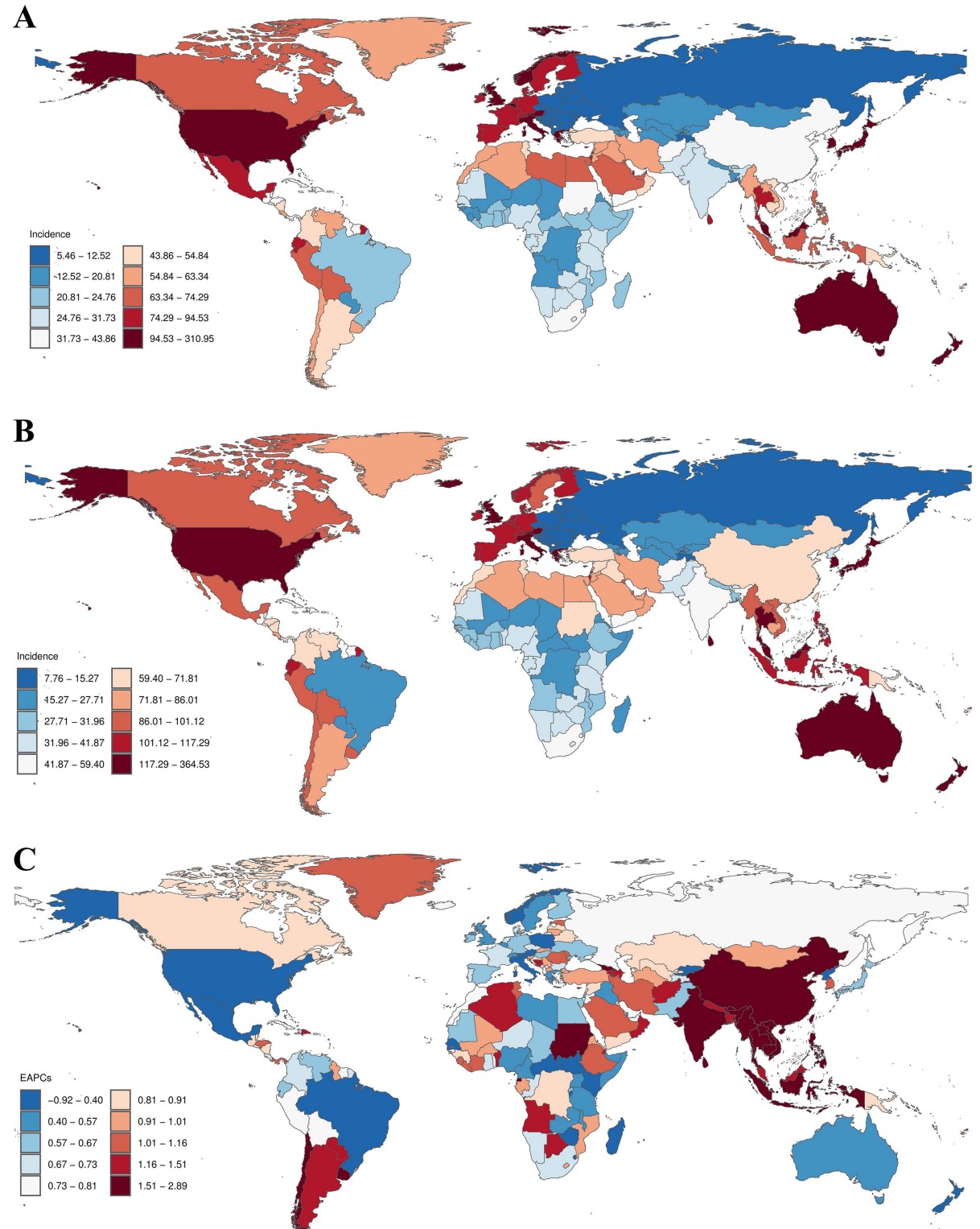

**Fig 2. Incidence of PCOS among women of reproductive age across 204 countries and territories between 1990 and 2021.** (A) ASIR in 1990. (B) ASIR in 2021. (C) EAPC between 1990 and 2021.

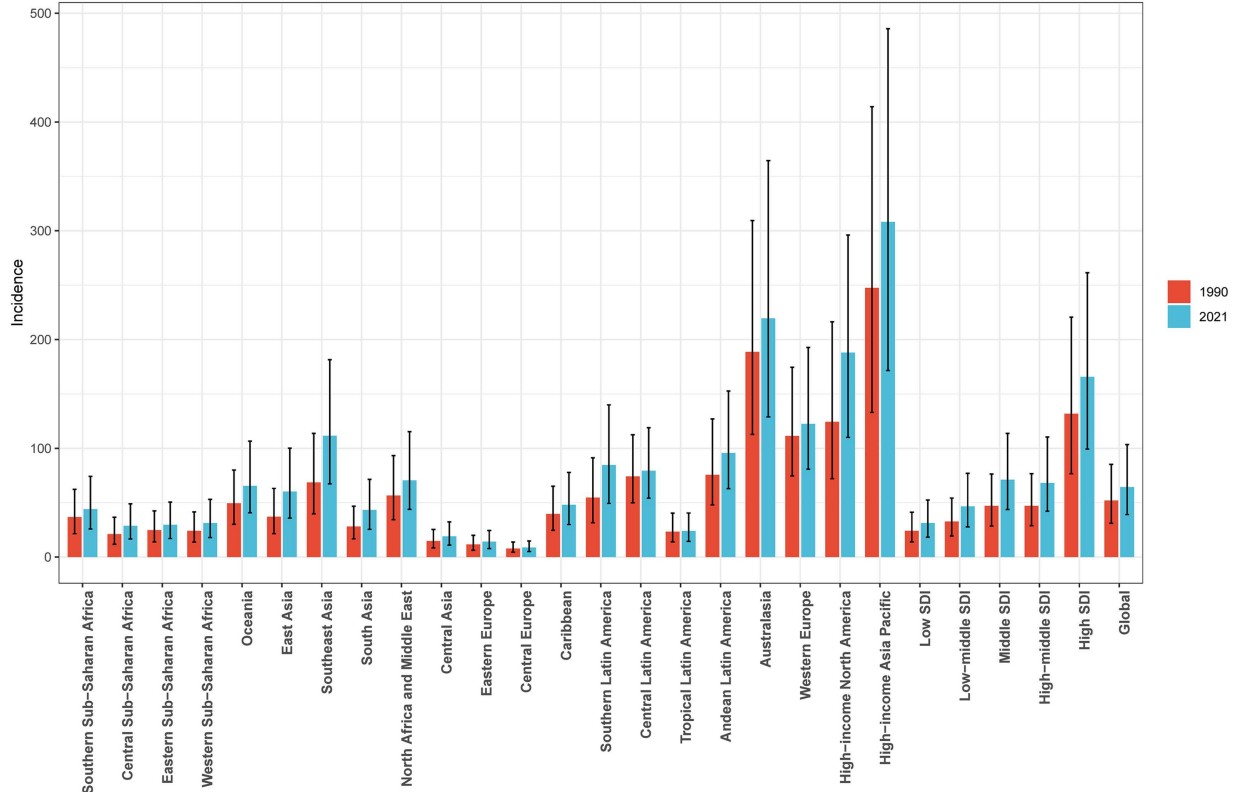

**Fig 3. The ASIR of PCOS among women of reproductive age at the global and regional level in 1990 and 2021.**

CI: 3.10, 3.67). In contrast, the United States of America showed the most significant decrease in ASPR, with an EAPC of −0.58 (95% CI: −1.12, −0.04). Globally, the ASPR of PCOS among women of reproductive age was estimated at 2628.48 (95%UI: 1870.21, 3668.82) per 100,000 population in 2021. This global ASPR exceeded that of 102 out of 204 countries, while remaining lower than the incidence rates observed in 102 countries (S4 Table).

**DALYs.** In 2021, People's Republic of China reported the highest absolute DALYs of PCOS globally, with an estimated 9481.52*10³ DALYs (95% UI: 6591.67*10³, 13488.89*10³). Conversely, Republic of Italy demonstrated the highest ASDR at 135.48 (95% UI: 59.84, 288.85) per 100,000 population. Longitudinal analysis from 1990 to 2021 revealed that the Republic of Maldives experienced the most substantial increase in ASDR, with an EAPC of 3.39 (95%CI: 3.11, 3.68). In contrast, the United States of America showed the most significant decrease in ASDR, with an EAPC of −0.58 (95% CI: −1.12, −0.04). Globally, the ASDR of PCOS among women of reproductive age was estimated at 23.16 (95% UI: 10.27,48.43) per 100,000 population in 2021. This global ASPR exceeded that of 102 out of 204 countries, while remaining lower than the incidence rates observed in 102 countries (S5 Table).

### Projection of the Global Burden of PCOS among women of reproductive age

We utilized the BAPC model to project the incidence, prevalence, and DALYs of PCOS among women of reproductive age. Our projections revealed a consistently increasing trend in incidence, prevalence and DALYs of PCOS in 2050. Based on the BAPC model, the incidence was projected to rise to 113.18 (95% CI: 34.80, 191.54) per 100,000 population by 2050, with the highest incidence was observed in the 15–20 age group at 65.52 (95% CI: 15.79, 115.25) per 100,000 population (S11 Fig). The prevalence was projected to rise to 4332.52 (95% CI: 3012.45, 5652.59) per 100,000 population

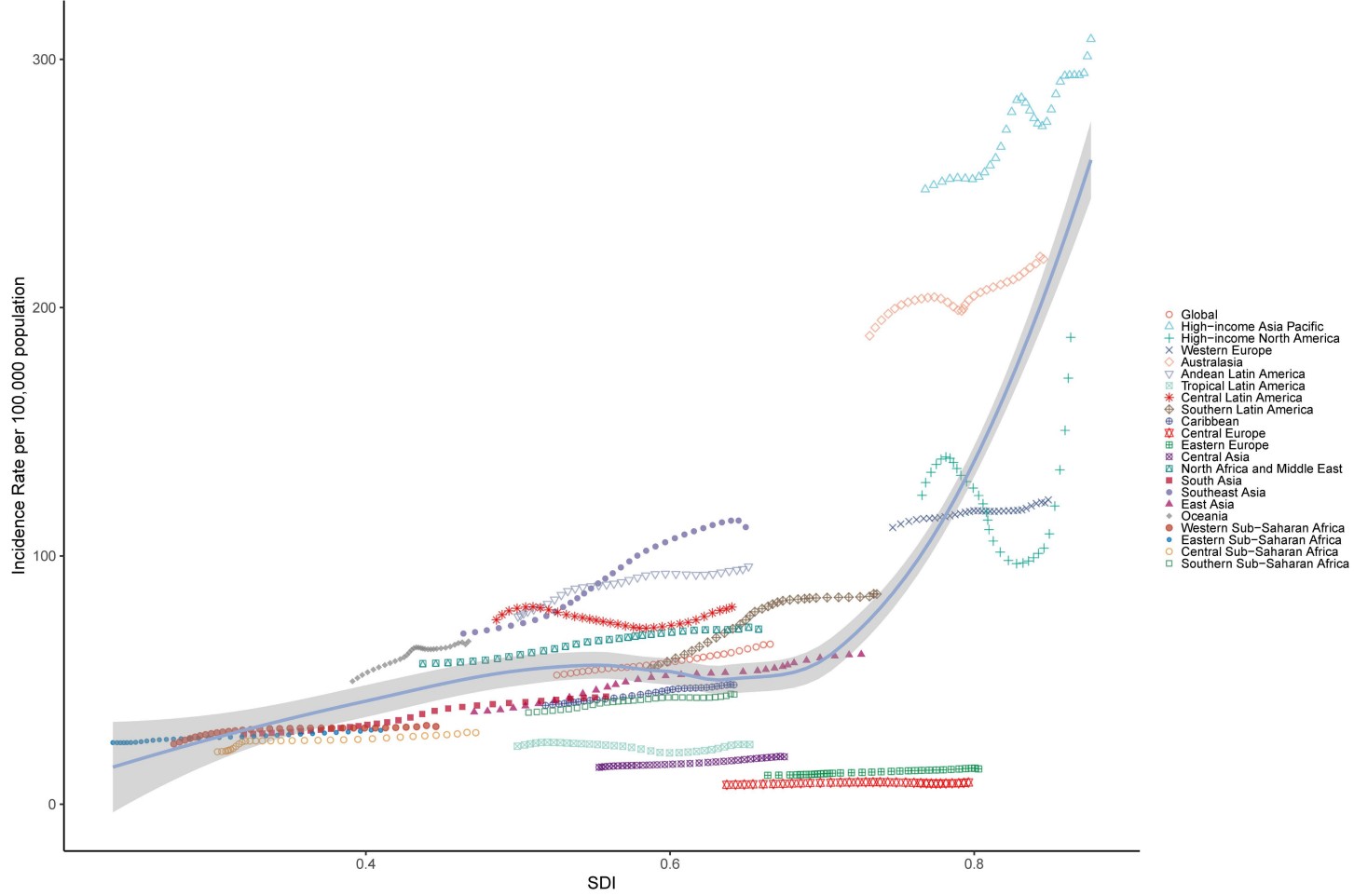

**Fig 4. Association between ASIR and SDI of PCOS among reproductive-aged women, 1990–2021.**

by 2050, with the highest prevalence observed in the 40–44 age group at 476.2 (95% CI: 434.16, 519.28) per 100,000 population (S12 Fig). The DALYs was projected to rise to 37.81 (95% CI: 25.81, 49.81) per 100,000 population by 2050, with the highest DALYs observed in the 20–24 age group at 4.19 (95% CI: 1.84, 6.55) per 100,000 population (S13 Fig).

### Frontier analysis of PCOS among women of reproductive age

Based on data spanning from 1990 to 2021, frontier analysis was performed to assess the potential improvement space in the DALYs for PCOS among women of reproductive age. This analysis incorporated key metrics, including the ASDR and the SDI, to explore variations in national and regional development levels. The 15 countries and territories demonstrating the greatest disparities in potential improvement in ASDR (effective difference range: 133.41–59.60) include Italy, Japan, New Zealand, Australia, Malaysia, United States of America, Austria, United Kingdom, Brunei Darussalam, Iceland, Mauritius, Singapore, Monaco, Luxembourg, and Greece. In contrast, the 15 countries and territories with the smallest disparities in potential improvement in ASDR (effective difference range: 1.12–2.01) include Bosnia and Herzegovina, Albania, North Macedonia, Serbia, Czechia, Somalia, Romania, Slovakia, Croatia, Hungary, Montenegro, Bulgaria, Slovenia, Burundi, and Ukraine (Fig 5).

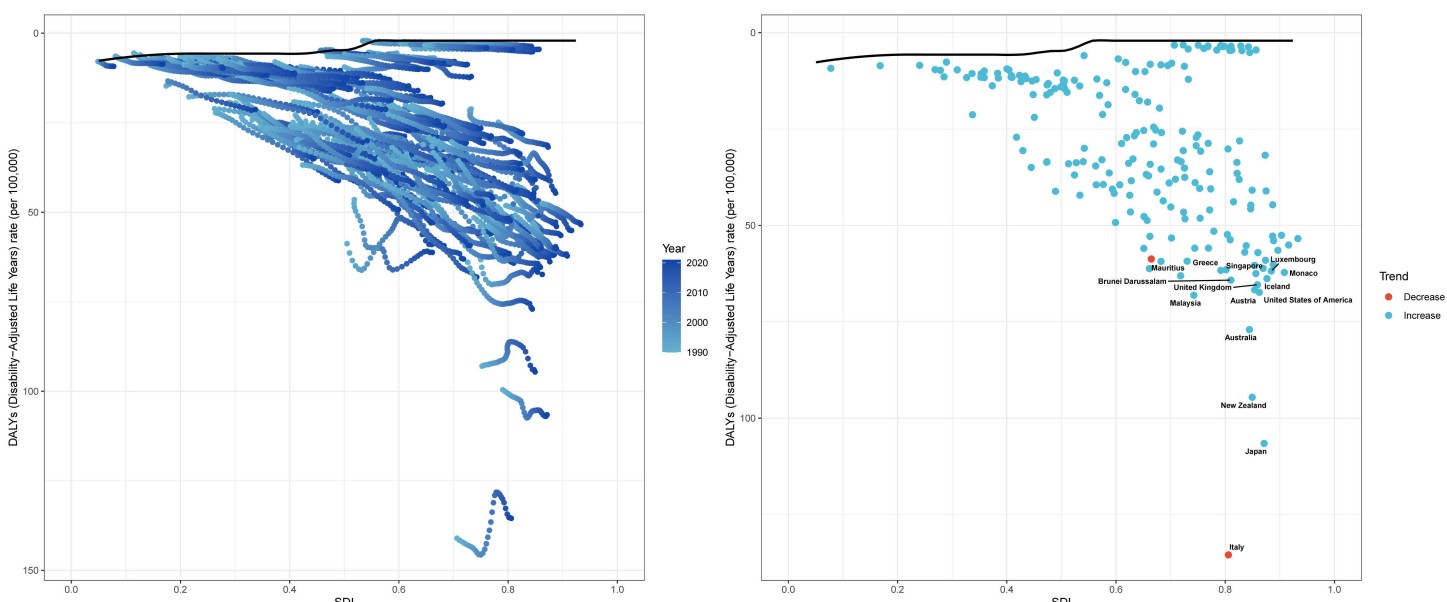

**Fig 5. Frontier analysis exploring the relationship between SDI and ASDR for PCOS among reproductive-aged women from 1990 to 2021.**

## Discussion

PCOS represents a significant global health burden, predominantly affecting women of reproductive age and exerting profound clinical implications across the lifespan, from adolescence to postmenopausal years [23,24]. This 31-year study, spanning historical data and projecting disease burden through 2050, encompasses comprehensive analyses of reproductive-aged women across 204 countries and territories, 21 GBD regions, and five SDI quintiles. Taking advantage of the innovative application of GBD 2021 data, this research represents the first attempt to systematically evaluate temporal trends and health disparities in the PCOS. The study identifies pivotal years with significant shifts in disease metrics and quantifies the impact of socioeconomic development levels on cross-national variations in disease burden. Traditional analytical approaches face limitations in addressing the nonlinear, multivariate, and high-dimensional characteristics of GBD 2021 data. To overcome these challenges, this study employs advanced analytical methodologies, ensuring robust and precise interpretation of complex datasets. These findings not only provide novel insights into the global burden of PCOS but also offer evidence-based recommendations to guide public health policymakers in dealing with this critical health issue.

The findings of this study demonstrated a consistent upward trend in the global burden of PCOS among women of reproductive age from 1990 to 2021, with significant percentage increases in ASIR (23.92%), ASPR (28.00%), and ASDR (27.42%). Our findings align with previous studies utilizing the GBD 2019 data, demonstrating consistency in the observed trends of PCOS burden [5]. Although direct comparisons are limited due to differences in temporal coverage and methodological approaches, the convergent conclusions from these independent analyses underscore the growing recognition of PCOS as a significant global public health challenge.

Significant age-related disparities were observed in the epidemiological characteristics of PCOS, highlighting distinct burden patterns across different age groups. In this study, women of reproductive age were categorized into seven 5-year age groups: 15–19, 20–24, 25–29, 30–34, 35–39, 40–44, and 45–49 years. The findings of this study indicate that the ASIR of PCOS among women aged 15–19 years is the highest across all age groups. Simultaneously, both the ASPR and ASDR were highest in the 20–24 age group. The slight variations in these results may be attributed to the challenges associated with diagnosing PCOS during adolescence, as the symptoms of PCOS often overlap with physiological

characteristics and changes commonly observed during puberty. Furthermore, differences in the selection and application of diagnostic criteria for PCOS across various national and regional settings, as well as the limited availability of diagnostic techniques such as transvaginal ultrasound in certain regions (e.g., the Middle East and North Africa), may also contribute to these discrepancies [15,25].

The exclusive adoption of NIH 1990 criteria in the GBD study may substantially underestimate the true burden of PCOS in regions where Rotterdam or AE-PCOS criteria are routinely applied. Extensive data indicate that diagnostic criteria critically determine PCOS prevalence estimates: NIH-based rates range 4%−6.6% globally, whereas Rotterdam criteria yield prevalences of 6%−21% [3]. This discrepancy primarily arises from Rotterdam's inclusion of broader phenotypes (e.g., normoandrogenic phenotypes with ovulatory dysfunction and polycystic ovarian morphology) [3,26]. Crucially, country-specific studies in Turkey and Denmark demonstrate Rotterdam-based prevalence rates reaching 2–3 times NIH-based figures (e.g., 19.9% vs 6.1% in Turkey) [27]. GBD's sole reliance on NIH criteria thus introduces dual biases: First, it systematically underestimates disease magnitude in regions predominantly using Rotterdam criteria. Second, it excludes large subgroups of mild or non-hyperandrogenic phenotypes, distorting public health resource allocation [28]. We recommend incorporating sensitivity analyses across diagnostic criteria through stratified burden reporting to better reflect global epidemiological variations.

This study also investigated the trends in the incidence, prevalence, and DALYs associated with PCOS, revealing distinct patterns across different SDI regions. In 2021, the high SDI region exhibited the highest ASIR, ASPR, and ASDR among the five SDI regions. Furthermore, the middle SDI region demonstrated the highest EAPC in ASIR, ASPR, and ASDR during the period from 1990 to 2021. Overall, the burden of PCOS among women of reproductive age across the 21 GBD regions exhibited fluctuations with increasing SDI levels. Specifically, the burden gradually increased from low SDI to middle SDI regions, showed a slight decrease from middle SDI to high-middle SDI regions, and then experienced a substantial rise from high-middle SDI to high SDI regions. These findings were consistent with some previous studies [5,9,13,14], yet they conflicted the conclusions of others [12,15]. This discrepancy may be attributed to the use of outdated data (GBD 2019 or earlier versions) in prior research or differences in the geographical regions examined. The elevated burden of PCOS observed in high SDI regions may be attributed to two primary factors: enhanced diagnostic capabilities within advanced healthcare systems and specific population characteristics. Firstly, the sophisticated healthcare infrastructure in these regions facilitates more accurate and timely detection of PCOS cases, potentially leading to higher reported prevalence rates. Secondly, the increased disease burden may be associated with modifiable risk factors prevalent in high SDI populations, including sedentary lifestyles, Westernized dietary patterns, and consequent metabolic disturbances such as insulin resistance. Furthermore, demographic shifts toward an aging population in these regions may contribute to the observed epidemiological patterns, as PCOS-related complications often manifest or persist into later reproductive years [29]. In contrast, regions with low SDI demonstrate a comparatively lower reported prevalence of PCOS, which may largely reflect systemic limitations in healthcare delivery rather than true epidemiological differences. This apparent reduction in disease burden is primarily attributable to critical gaps in healthcare infrastructure, including inadequate diagnostic facilities and limited availability of specialized reproductive health services. Furthermore, the lack of standardized screening protocols and restricted access to advanced diagnostic modalities, such as ultrasonography and hormonal assays, likely contribute to significant underdiagnosis and underreporting of PCOS cases in these resource-constrained settings [30].

These findings underscore the need for tailored policy interventions across SDI regions to address the varying PCOS burden effectively. In high-SDI regions, policies should prioritize comprehensive PCOS screening programs integrated into routine reproductive healthcare, coupled with public health campaigns targeting modifiable risk factors such as physical inactivity and metabolic disorders through workplace wellness initiatives and nutritional education. For middle-SDI regions, where the most rapid increase in burden was observed, health systems should focus on capacity-building for early diagnosis through primary care physician training and standardized diagnostic protocols, while investing in cost-effective interventions such as community-based lifestyle modification programs. Low-SDI regions would benefit most from foundational healthcare strengthening, including the deployment of simplified diagnostic tools (e.g., mobile ultrasound units) and

task-shifting strategies to enable frontline healthcare workers to identify PCOS cases. Across all SDI levels, longitudinal research funding should be allocated to clarify the interplay between socioeconomic development and PCOS epidemiology, with particular attention to distinguishing healthcare access limitations from true biological risk factors in observed disparities, as this will be critical for refining global guidelines and ensuring equitable resource allocation aligned with the UN's Sustainable Development Goals for women's health.

In this study, we employed frontier analysis to systematically evaluate the potential for reducing the global burden of PCOS across 204 countries and territories. Building upon previous GBD studies related to PCOS, this represents the first application of frontier analysis modeling to quantify the improvement potential in PCOS management and outcomes. This innovative analytical approach enables the identification of best-performing regions and establishes achievable benchmarks for disease care, providing valuable insights for health system optimization and resource allocation strategies [31]. Our frontier analysis revealed a paradoxical relationship between socioeconomic development and PCOS-related health outcomes. While PCOS-associated health loss demonstrated significant associations with socioeconomic and demographic indicators, the analysis identified several developing countries achieving superior outcomes in PCOS-related DALYs. These high-performing developing nations may provide valuable insights and potential best-practice models for developed countries seeking to optimize their PCOS management strategies, challenging conventional assumptions about the relationship between economic development and health system performance in chronic endocrine disorders. The escalating burden of PCOS posed significant threats to women's quality of life and reproductive health worldwide [32–34]. Given its profound implications, strategic interventions targeting PCOS risk factor reduction should be prioritized in both global and national health agendas. Such proactive measures are essential for mitigating disease prevalence across diverse populations and minimizing the risk of PCOS-related complications, thereby addressing this growing public health challenge through comprehensive, evidence-based approaches [35].

This study projected the burden of PCOS among women of reproductive age through 2050, revealing a concerning upward trajectory in PCOS-related health impacts during the period from 2021 to 2050. The predictive modeling revealed a consistent year-on-year increase in the global burden of PCOS, underscoring the urgent need for targeted public health interventions to address this escalating challenge in women's health. The results of this study demonstrate a consistent predictive trend with previous research, despite methodological differences in population selection [11]. While our analysis specifically targeted women of reproductive age, the earlier study included female populations across all age groups. This convergence of findings across distinct demographic strata not only reinforces the validity of our predictive models but also suggests a robust and generalizable pattern in the projected burden of PCOS, supporting the reliability and clinical relevance of these epidemiological projections.

Looking ahead to 2050 when PCOS burden is projected to increase substantially, critical knowledge gaps remain regarding optimal intervention strategies, particularly concerning phenotype-specific approaches and scalable solutions for diverse populations; future studies should therefore prioritize three key areas: (1) large-scale randomized trials comparing lifestyle modifications, pharmacotherapy, and their combinations stratified by Rotterdam phenotypes, (2) cost-effectiveness analyses of early intervention in adolescent populations showing early PCOS markers, and (3) implementation research testing digital health platforms for long-term management in low-resource settings, as these approaches would not only address current evidence limitations but also provide actionable data for healthcare systems preparing for the anticipated rise in PCOS prevalence.

This study has several limitations that warrant careful consideration. First, the reliance on the DisMod-MR model within the GBD framework, which exclusively adopts the NIH diagnostic criteria for PCOS, may introduce heterogeneity in epidemiological estimates due to the exclusion of alternative diagnostic standards [36]. Second, while the GBD 2021 dataset provides comprehensive global coverage, inconsistencies in data reporting across countries and territories could affect the precision of our findings. Finally, discrepancies between GBD estimates and real-world data, particularly in low- and middle-income regions, may lead to underestimations due to limited medical resources and inconsistent reporting in certain regions, which could impact the accuracy of global burden estimates. These limitations highlight the need for improved data quality and standardized reporting to enhance the reliability of future epidemiological projections.

## Conclusion

PCOS represents one of the most prevalent endocrine disorders among women of reproductive age, contributing significantly to the global burden of disease. The escalating burden of PCOS necessitates urgent attention from the global health community. Health management authorities require robust epidemiological data to facilitate effective monitoring and intervention strategies within their jurisdictions to address this growing public health challenge. This study provides critical epidemiological insights that can serve as a valuable reference for health policymakers and management institutions in developing targeted strategies to mitigate the widespread impact of PCOS.

## Supporting information

**S1 Table. Prevalence of PCOS among women of reproductive age between 1990 and 2021 at the global and regional level.**
(DOCX)

**S2 Table. DALYs of PCOS among women of reproductive age between 1990 and 2021 at the global and regional level.**
(DOCX)

**S1 Fig. Age-stratified percentages of PCOS incidence among women of reproductive age at the global and regional level in 1990 and 2021.**
(TIF)

**S2 Fig. Prevalence of PCOS among women of reproductive age across 204 countries and territories between 1990 and 2021.** (A) ASPR in 1990. (B) ASPR in 2021. (C) EAPC between 1990 and 2021.
(TIF)

**S3 Fig. Age-stratified percentages of PCOS prevalence among women of reproductive age at the global and regional level in 1990 and 2021.**
(TIF)

**S4 Fig. DALYs of PCOS among women of reproductive age across 204 countries and territories between 1990 and 2021.** (A) ASDR in 1990. (B) ASDR in 2021. (C) EAPC between 1990 and 2021.
(TIF)

**S5 Fig. Age-stratified percentages of PCOS DALYs among women of reproductive age at the global and regional level in 1990 and 2021.**
(TIF)

**S6 Fig. The ASPR of PCOS among women of reproductive age at the global and regional level in 1990 and 2021.**
(TIF)

**S7 Fig. The ASDR of PCOS among women of reproductive age at the global and regional level in 1990 and 2021.**
(TIF)

**S8 Fig. The association between EAPC and ASRs of PCOS among women of reproductive age (left) and the association between EAPC and SDI (right) in 2021.** (A) ASIR. (B) ASPR. (C) ASDR.
(TIF)

**S9 Fig. Association between ASPR and SDI of PCOS among reproductive-aged women, 1990–2021.**
(TIF)

**S10 Fig.** Association between ASDR and SDI of PCOS among reproductive-aged women, 1990–2021.
(TIF)

**S3 Table.** Incidence of PCOS among women of reproductive age between 1990 and 2021 at the national level.
(DOCX)

**S4 Table.** Prevalence of PCOS among women of reproductive age between 1990 and 2021 at the national level.
(DOCX)

**S5 Table.** DALYs of PCOS among women of reproductive age between 1990 and 2021 at the national level.
(DOCX)

**S11 Fig.** Predict the incidence of PCOS among reproductive-aged women in 2050.
(TIF)

**S12 Fig.** Predict the prevalence of PCOS among reproductive-aged women in 2050.
(TIF)

**S13 Fig.** Predict the DALYs of PCOS among reproductive-aged women in 2050.
(TIF)

## Acknowledgments

We would like to express our sincere gratitude to the staff and collaborators of the Institute for Health Metrics and Evaluation (IHME) for their efforts in preparing and providing these publicly available data. Additionally, we are deeply grateful to the Central Hospital of Dalian University of Technology for their substantial support throughout this study.

## Author contributions

**Conceptualization:** Xinkuo Zheng, Yalin Xi.

**Data curation:** Xinkuo Zheng, Meishen Liu, Zhaohui Bai, Lili Geng, Yun Shen, Jing Na.

**Formal analysis:** Xinkuo Zheng.

**Investigation:** Ling Wu, Yun Shen, Yalin Xi.

**Methodology:** Jing Na, Haonan Qiu.

**Software:** Meishen Liu, Haonan Qiu.

**Validation:** Zhaohui Bai.

**Writing – original draft:** Zhaohui Bai.

**Writing – review & editing:** Xinkuo Zheng.

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
