## [Decision Letter · Decision Letter 0]

11 Jul 2025

PONE-D-25-17138Global burden of Polycystic Ovary Syndrome in women of reproductive age, 1990-2021: Analysis of the Global Burden of Disease Study 2021 with projections to 2050PLOS ONE

Dear Dr. Zheng,

Thank you for submitting your manuscript to PLOS ONE. After careful consideration, we feel that it has merit but does not fully meet PLOS ONE’s publication criteria as it currently stands. Therefore, we invite you to submit a revised version of the manuscript that addresses the points raised during the review process.

**ACADEMIC EDITOR:** All the two reviewers recommended publication of your manuscript after minor revision.  Please treat all the comments seriously. One of the reviewers mentioned "Software Concerns". This should be resolved. 

We look forward to receiving your revised manuscript.

Kind regards,

Wan-Xi Yang, Ph.D.

Academic Editor

PLOS ONE

Journal Requirements:

3. We note that of Figure 2 , S2 & S4 Figure in your submission contain [map/satellite] images which may be copyrighted. All PLOS content is published under the Creative Commons Attribution License (CC BY 4.0), which means that the manuscript, images, and Supporting Information files will be freely available online, and any third party is permitted to access, download, copy, distribute, and use these materials in any way, even commercially, with proper attribution. For these reasons, we cannot publish previously copyrighted maps or satellite images created using proprietary data, such as Google software (Google Maps, Street View, and Earth). For more information, see our copyright guidelines: http://journals.plos.org/plosone/s/licenses-and-copyright.

a. You may seek permission from the original copyright holder of Figure 2 , S2 & S4 Figure to publish the content specifically under the CC BY 4.0 license. 

In the figure caption of the copyrighted figure, please include the following text: “Reprinted from [ref] under a CC BY license, with permission from [name of publisher], original copyright [original copyright year].

Reviewers' comments:

Reviewer's Responses to Questions

**Comments to the Author**

1. Is the manuscript technically sound, and do the data support the conclusions?

Reviewer #1: Yes

Reviewer #2: Yes

2. Has the statistical analysis been performed appropriately and rigorously? 

Reviewer #1: Yes

Reviewer #2: Yes

3. Have the authors made all data underlying the findings in their manuscript fully available?

Reviewer #1: Yes

Reviewer #2: Yes

4. Is the manuscript presented in an intelligible fashion and written in standard English?

Reviewer #1: Yes

Reviewer #2: Yes

5. Review Comments to the Author

Reviewer #1: Reviewer Comments

Title: Global burden of Polycystic Ovary Syndrome in women of reproductive age, 1990-2021: Analysis of the Global Burden of Disease Study 2021 with projections to 2050

Date: April 18, 2025

Recommendation: Minor Revision

Summary: This manuscript conduct analysis of the global burden of Polycystic Ovary Syndrome in reproductive-aged women using GBD 2021 data. Findings reveal consistently increasing incidence, prevalence, and disability-adjusted life years worldwide, with significant regional variations.

Comments:

Software Concerns: The manuscript states that "All statistical analyses and visualizations were performed using R software (version 4.4.2) and JD_GBDR (V2.36, Jingding Medical Technology Co., Ltd)". Please clarify the specific role of the JD_GBDR software in the analysis. Consider to provide sufficient detail for transparency or confirm if the analyses can be fully replicated using only R and its standard packages.

Figure Clarity: In the reviewed PDF version, the legibility of some figures, notably Figure 5, is suboptimal. Please ensure that all figures are provided at a sufficiently high resolution in the final submission.

Reviewer #2: Abstract: adequate, well-written, suggest to spell out EAPC in the abstract.

Introduction: well-written & adequate

Methodology: Well-written & adequate

Results: Well-written, comprehensive analysis, clear images used, clear explanation provided

Discussion:

well-writted & adequate.

- Line 281: repetition of “but also”

- suggest to discuss the impact of exclusive used of the NIH criteria in GBD that might affect the actual incidence & prevalence of PCOS in countries where Rotterdam or AE-PCOS criteria are used.

- The manuscript mentioned the impact of non-modifiable risk factors such as sedentary lifestyles and insulin resistance. suggest to support this using quantitative linkage or risk attribution analysis

- with the projected increased burden of PCOS by the year 2050, suggest to add future studies that could evaluate intervention effectiveness.

- Suggest potential policy level recommendations based on the different SDI regions.

Conclusion: Good conclusion provided.

6. PLOS authors have the option to publish the peer review history of their article (what does this mean? ). If published, this will include your full peer review and any attached files.

**Do you want your identity to be public for this peer review?** For information about this choice, including consent withdrawal, please see our Privacy Policy .

Reviewer #1: No

Reviewer #2: **Yes: ** INTAN SUHANA ZULKAFLI

---

## [Decision Letter · Decision Letter 1]

9 Sep 2025

Global burden of polycystic ovary syndrome in women of reproductive age, 1990-2021: Analysis of the global burden of disease study 2021 with projections to 2050

PONE-D-25-17138R1

Dear Dr. Zheng,

We’re pleased to inform you that your manuscript has been judged scientifically suitable for publication and will be formally accepted for publication once it meets all outstanding technical requirements.

Kind regards,

Wan-Xi Yang, Ph.D.

Academic Editor

PLOS ONE

Additional Editor Comments (optional):

Reviewer #1:

Reviewer #2:

Reviewers' comments:

Reviewer's Responses to Questions

**Comments to the Author**

1. If the authors have adequately addressed your comments raised in a previous round of review and you feel that this manuscript is now acceptable for publication, you may indicate that here to bypass the “Comments to the Author” section, enter your conflict of interest statement in the “Confidential to Editor” section, and submit your "Accept" recommendation.

Reviewer #1: All comments have been addressed

Reviewer #2: All comments have been addressed

2. Is the manuscript technically sound, and do the data support the conclusions?

Reviewer #1: Yes

Reviewer #2: Yes

3. Has the statistical analysis been performed appropriately and rigorously? 

Reviewer #1: Yes

Reviewer #2: Yes

4. Have the authors made all data underlying the findings in their manuscript fully available?

Reviewer #1: Yes

Reviewer #2: Yes

5. Is the manuscript presented in an intelligible fashion and written in standard English?

Reviewer #1: Yes

Reviewer #2: Yes

6. Review Comments to the Author

Reviewer #1: (No Response)

Reviewer #2: The authors have addressed all comments satisfactorily and as a result, the manuscript has improved greatly. Congratulations to the team for the good work.

7. PLOS authors have the option to publish the peer review history of their article (what does this mean? ). If published, this will include your full peer review and any attached files.

**Do you want your identity to be public for this peer review?** For information about this choice, including consent withdrawal, please see our Privacy Policy .

Reviewer #1: No

Reviewer #2: **Yes: ** INTAN SUHANA ZULKAFLI

---

## [Editor Report · Acceptance letter]

PONE-D-25-17138R1

PLOS ONE

Dear Dr. Zheng,

I'm pleased to inform you that your manuscript has been deemed suitable for publication in PLOS ONE. Congratulations! Your manuscript is now being handed over to our production team.

Kind regards,

on behalf of

Dr. Wan-Xi Yang

Academic Editor

PLOS ONE